


# Coupling Human and Natural Systems for Sustainability: Experiences from China's Loess Plateau

Bojie Fu[1,2], Xutong Wu[2], Zhuangzhuang Wang[1], Xilin Wu[1], Shuai Wang[2]

[1]State Key Laboratory of Urban and Regional Ecology, Research Center for Eco-Environmental Sciences, Chinese Academy
of Sciences, Beijing 100085, China
[2]State Key Laboratory of Earth Surface Processes and Resource Ecology, Faculty of Geographical Science, Beijing Normal
University, Beijing 100875, China

*Correspondence to*: Bojie Fu (bfu@rcees.ac.cn)

**Abstract.** Addressing the sustainability challenges facing humanity in the Anthropocene requires the coupling of human and
natural systems, rather than their separate treatment. To understand the dynamics of a coupled human and natural system
(CHANS) and promote its sustainability, we proposed a conceptual cascade framework of "Pattern-Process-Service-
Sustainability". The use of this framework was systematically illustrated by a review of CHANS research experiences in
China's Loess Plateau (LP) in terms of coupling landscape patterns and ecological processes, linking ecological processes to
services, and promoting social-ecological sustainability. The LP is well-known for its historically notorious soil erosion and
successful vegetation restoration achieved in recent decades. Vegetation coverage in the LP has increased since 2000 due to
ecological restoration. Soil erosion has been well controlled and the sediment deriving from the LP, and flowing into the
Yellow River, has greatly decreased; however, overplanting, the introduction of exotic plant species, and the mismanagement
of planted vegetation have also led to soil drying in some areas. Ecosystem services, especially for soil conservation and carbon
sequestration, have significantly improved, although a trade-off between carbon sequestration and water supply has been
identified at multiple scales. Based on the comprehensive understanding of CHANS dynamics, targeted policy and
management suggestions are here proposed to support the social-ecological sustainability of the LP. The research experience
accumulated on the LP offers examples of the application of the "Pattern-Process-Service-Sustainability" framework. Future
research using this framework should especially examine the integrated research of multiple processes, the cascades of
ecosystem structure, function, services, and human-wellbeing, the feedback mechanisms of human and natural systems, and
the data and models for sustainability.

## 1 Introduction

In the Anthropocene, accelerated landscape modification and climate change driven by human activities are altering the earth's
surface processes, leading to a series of social and environmental issues (Steffen et al., 2007; Lewis and Maslin, 2015). Climate
change, forest loss, ecosystem degradation, and the depletion of resources are global concerns, threatening the sustainability
of human society (IPBES, 2019; MA, 2005). Addressing human society toward a path of sustainable development has become



an urgent item on the research agenda of the scientific community and policy makers (UN, 2015). To promote the international cooperation on sustainable development, in 2000 the UN millennium declaration agreed on the eight Millennium Development Goals, and in 2015 the launch of the 2030 Agenda by the UN announced the 17 Sustainable Development Goals (SDGs). However, recent reports suggested that it may not be possible to achieve the SDGs by 2030 (UN, 2019), mainly due to the

complex and intertwined nature of these pressing sustainability challenges that human beings face (Biggs et al., 2021). In recent decades, a growing consensus in academia has been reached, asserting that sustainability challenges cannot be properly addressed until humans and nature are treated as an entirety (Biggs et al., 2021; Cumming, 2014).

The coupled human and natural systems (CHANSs), which are also referred to as coupled human and earth systems, human and environment systems, and social-ecological systems, describe a complex phenomenon in which human processes and

natural processes interact and co-evolve (Wang et al., 2018b; Quintas-Soriano et al., 2021; Ostrom, 2009; Liu et al., 2009; Fu and Wei, 2018; Cumming, 2014). In a natural system, the atmosphere, hydrosphere, biosphere, and solid earth interact with each other, constantly exchanging materials, energy, and information (DeFries, 2008). Humans in the biosphere strive to survive and develop through the modification of ecosystem patterns and processes (DeFries, 2008). Therefore, CHANSs use a holistic perspective to integrate patterns and processes that connect human and natural systems, as well as within-scale and

cross-scale interactions and feedbacks between them (Liu et al., 2021). In addition, these concepts provide guidance on how to combine the human and natural dimensions within an integrated framework, and elucidate their complex feedback mechanisms (Liu et al., 2013a; Tian, 2017; Partelow, 2018). Such an integrated framework is needed to develop innovative insights and solutions to the sustainability challenges that characterize the Anthropocene (Liu et al., 2021).

In CHANSs, humans alter the ecosystem patterns and processes to obtain desired ecosystem services (ESs), such as food,

freshwater, and timber (Liu et al., 2009). Here, ESs represent the benefits that people directly or indirectly obtain from ecosystems, and their changes will affect human well-being and sustainable development (Costanza et al., 2017). The modification of ecosystems driven by human demands and preferences often unconsciously leads to declines in some other services, especially for regulating and supporting services, causing a series of social and environmental problems, which in turn threaten the sustainability of CHANSs (Liu et al., 2009; Bennett et al., 2009). Pattern-process coupling provides a critical

approach for integrated research and a comprehensive understanding of the mechanisms of natural processes, such as the carbon-water cycle, ecohydrological processes, and soil-vegetation-atmosphere interactions (Fu et al., 2019). Ecological processes and functions transform to services when they can directly and indirectly benefit humans (Costanza et al., 2017). Unlike the research on pattern-process coupling, ES research places a greater emphasis on the social dimension or human need of ecosystems, constituting an important bridge between science and policy. Managing ecosystem processes to sustain the

provision of ESs is the basic way to improve sustainability (Fu et al., 2013). For example, through land use planning and management, the land use structure can be optimized to obtain the desired and sustainable provision of ESs. To elucidate the complex feedback mechanism of CHANSs, it is necessary to couple ecosystem patterns and processes, and link ecosystem processes and services (Wang et al., 2018b), and the results obtained could support policy-related decisions that enhance sustainability.



To better observe, analyze, and predict the dynamics of CHANSs, we proposed the conceptual cascade framework of "Pattern-Process-Service-Sustainability" (Fu and Wei, 2018). First, the proposed framework and its components were introduced in detail, then its use to understand the dynamics of CHANSs and support decision making for the promotion of sustainability was systematically illustrated through a review of CHANS research conducted on China's Loess Plateau (LP). The LP is well-known for its historically notorious soil erosion and successful vegetation restoration achieved in recent decades (Wu et al.,

2020). Over the last 20 years, researchers have conducted extensive studies on the CHANS of the LP using the "Pattern-Process-Service-Sustainability" framework, making it an ideal region for summarizing research experiences of CHANS.

## 2 The Pattern-Process-Service-Sustainability framework

The proposed "Pattern-Process-Service-Sustainability" framework (Figure 1) provides a paradigm for researchers to study CHANSs and identify practical solutions for the adaptation to environmental change (Fu and Wei, 2018). The earth's surface

landscape has been extensively shaped and modified by natural forces and human activities, which are reflected in the pattern of landscape components (Steffen et al., 2015; Steffen et al., 2007). In this framework, the term "pattern" generally refers to a spatial pattern of landscape components, including their properties of size, type, number, and spatial distribution; and the term "process" indicates the dynamic features of the occurrence and evolution of incidents or phenomena (Fig. 1b) (Fu et al., 2011a). Patterns and processes interact with each other, which can be expressed as "patterns influence processes, and processes change

patterns" (Fu et al., 2011a; Fu et al., 2019). For example, land use configurations can influence soil erosion processes; and hydrological processes, such as floods, can quickly alter the land-cover pattern of alluvial plains. These reciprocal effects between patterns and processes are critical to understand the coupling mechanisms of CHANSs (Fu and Wei, 2018).

ESs are defined as the ecosystem characteristics, processes or functions that directly or indirectly contribute to human wellbeing (Costanza et al., 2017). A better understanding of ESs, and especially of the links between them and ecosystem

processes, is crucial to accurately assess and predict ES dynamics. The relationships between ecosystem processes and ESs are complex: multiple ESs may share the same ecological process, or multiple ecological processes may support the same single ES (Bennett et al., 2009; Fu et al., 2013). In addition, ecological processes interact with each other, rather than exist independently (Fu et al., 2015). For instance, both soil conservation and water yield services are affected by rainfall events, runoff, and vegetation dynamics. Therefore, changes in ecological patterns and processes caused by both natural forces and

human activities can affect multiple ecosystem services through cascade effects, resulting in trade-off and synergy effects between services (Fu et al., 2013). Specifically, a trade-off occurs when one service increases and another one decreases; while synergy indicates the presence of the same variation trend for both services (Bennett et al., 2009). Only when ecological processes and ESs are considered together, it is possible to comprehensively understand the driving mechanisms of ESs, and to identify the critical processes that could be regulated to optimize them (Fu et al., 2013).

A deep understanding of "Pattern-Process-Service" interactions represents the scientific basis for designing policies or measures that promote the "Sustainability" of CHANSs. Both environmental changes and human activities can affect the





interactions between landscape patterns and ecosystem processes, and directly or indirectly affect the ESs upon which humans depend (Fu et al., 2019). Among these factors, land use optimization is a feasible and effective way to improve the sustainability of CHANSs, providing an important opportunity to bridge science and policy. Thus, the above-mentioned

analytical paradigm for practical management can be integrated into a cascade framework of "Pattern-Process-Service-Sustainability" (Fu and Wei, 2018). To use this paradigm for the promotion of CHANS sustainability, it is necessary to couple landscape patterns and ecological processes, link ecological processes to services, and bridge science and policy (Fu et al., 2015). In the following sections, research conducted in China's LP was used as an example to illustrate how this framework is applied to empirical studies.

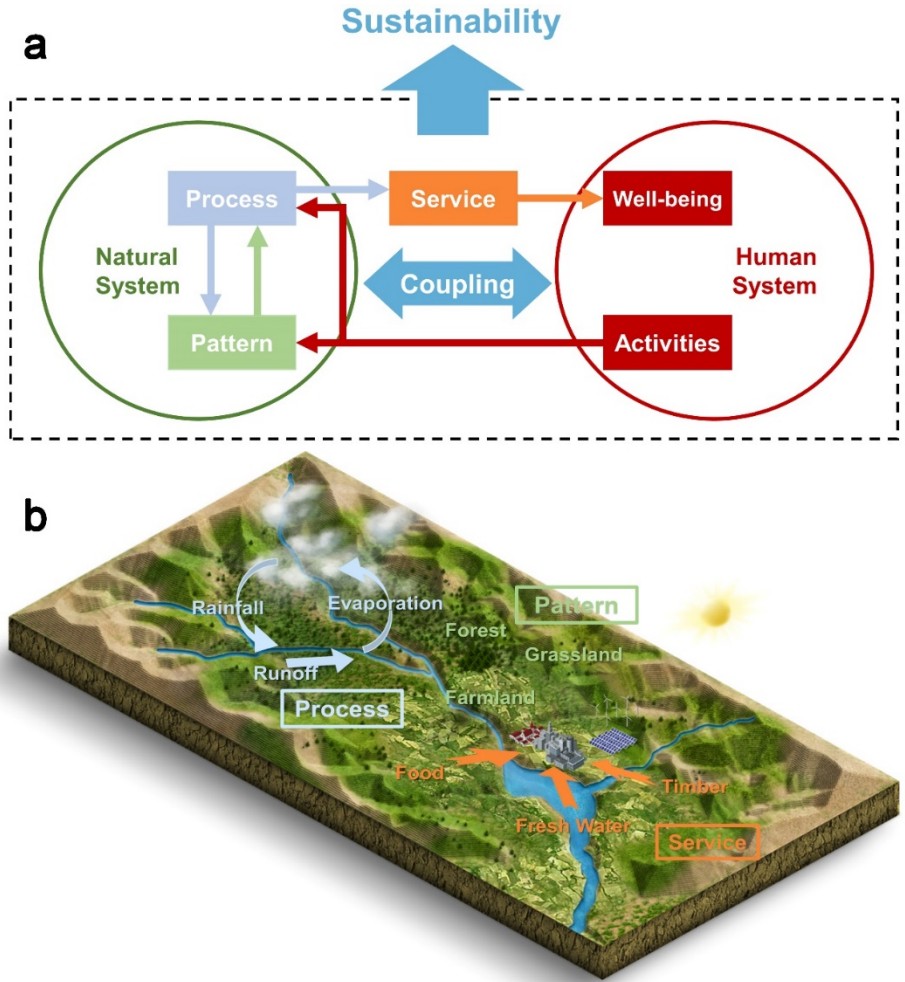


**Figure 1: Illustration of the "Pattern-Process-Service-Sustainability" framework. (a)** Pattern, process, service, and sustainability in the coupled human and natural system. **(b)** Diagram of typical patterns, processes (e.g., hydrological processes) and ESs (e.g., provision services).



## 3 China's Loess Plateau

The LP (Figure 2), located in central China, is the largest and deepest loess deposit in the world, covering an area of 640,000 km$^2$ (Fu et al., 2017). It is characterized by a continental monsoon climate with an average annual precipitation of 400 mm, and most of the plateau is located in a semiarid zone, based on the aridity index (Chen et al., 2007). Because the loess is highly fertile and easy to cultivate (Wang et al., 2010), early Chinese agriculture and civilization developed around the LP since the Neolithic era (Barton et al., 2009; Sun et al., 2018). However, due to the anthropic impacts affecting the area for thousands of

years, the LP has suffered from soil erosion, land degradation, natural disasters, and extreme poverty (Fu et al., 2017). These undesired social and ecological outcomes resulted from the mutually reinforcing feedbacks between natural and human processes (Wu et al., 2020).

The erosion susceptibility of loess, periodic extreme rainfall events, sparse vegetation, long-term and intensive farming activities shaped the fragile ecological environment on the LP, exacerbating the sensitivity of the region to extreme climatic

events and human disturbances (Fu et al., 2017). As the human population increased, more deforestation and land reclamation occurred to meet the growing demand for food and firewood. Long-term human-induced vegetation degradation also exacerbated soil erosion, turning more than 70% of the once-flat plateau into a region dominated by hills and gullies (Zhao et al., 2013). In addition, the Yellow River, the second largest river in China, flows through the LP, and receives nearly 90 % of the sediment load from this region (Wang et al., 2016). Due to the large amount of sediment deposited in the lower reaches of

the river, frequent flood and breach events have historically occurred, causing huge losses to the residents living in the alluvial plain (Chen et al., 2012; Chen, 2019). Furthermore, severe soil erosion caused a decline in land productivity and grain yield, and therefore more land needed to be reclaimed to feed the growing population (Chen et al., 2007). This vicious cycle seems to have determined the "fall" of the LP into a social-ecological trap.

To break out of this negative cycle, the Chinese government has taken numerous efforts to restore vegetation and control soil

erosion in the region, in order to establish a sustainable development path. Since the 1970s, numerous soil- and water-conservation measures have been implemented, including the construction of terracing and check-dams, integrated watershed management for soil and water conservation, and vegetation restoration (Fu et al., 2017). Among them, the Grain-to-Green Program (GTGP), implemented in 1999 in the LP, was the largest and most successful (Feng et al., 2016; Wu et al., 2020). Since the beginning of the 21$^{st}$ century, the vegetation coverage in the LP has significantly increased and soil erosion has been

effectively controlled (Fu et al., 2011b). Despite the success of the GTGP, many uncertainties remain in relation to the effects of ecological restoration on social-ecological sustainability (Fu et al., 2017), whose achievement in the LP has become the focus for researchers and policy-makers.

In the last two decades, extensive studies have been conducted on the CHANS of the LP to better understand its multiple natural and human processes, and to support policy measures for ecological restoration (Feng et al., 2010; Fu et al., 2011a; Fu

et al., 2017). The representative social-ecological context of the LP and the extensive research here conducted make it an ideal





region for summarizing research experiences of CHANS. In the following section, a systematic review of CHANS studies conducted on the LP using the "Pattern-Process-Service-Sustainability" framework is reported.

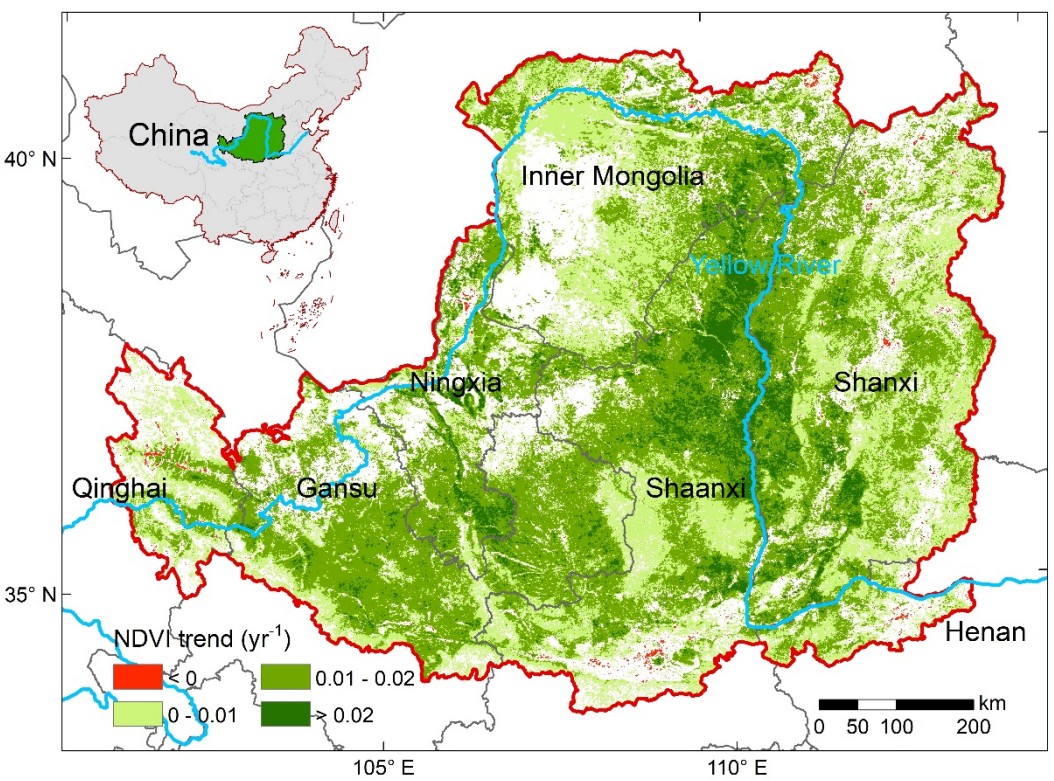

**Figure 2: Location of the LP and its normalized-difference vegetation index (NDVI) trend from 2000 to 2015.**

**4 Application of the framework in the LP**

**4.1 Coupling patterns and processes**

Through a process of natural evolution, the atmosphere-soil-vegetation interaction reaches a dynamic equilibrium, forming a relatively stable vegetation pattern (Zhang et al., 2018). As human disturbances to vegetation, deforestation and afforestation can alter this equilibrium and cause a series of ecohydrological effects that influence water, carbon, soil processes, and the

overall ecosystem sustainability (Feng et al., 2016; Zhang et al., 2018). The GTGP, launched in 1999, converted slope croplands into forest or grasslands, thus altering the vegetation cover in the LP (Fu et al., 2011b). Therefore, understanding the ecohydrological effects caused by vegetation restoration through the coupling of patterns and processes is of great significance to direct subsequent ecological restoration measures (Fu et al., 2011a). In order to achieve such understanding, researchers have conducted numerous studies on soil erosion processes, carbon-water process (Feng et al., 2016), and flow-

sediment relationships in the LP (Wang et al., 2016).



In the context of ecological restoration, the pressing task is to understand the soil erosion dynamics in the LP, and what land use pattern will result in the best soil and water conservation effect. Based on long-term field monitoring and control experiments (such as the building of runoff plots), studies have found that rainfall events characterized by high intensity, short duration, and high frequency had the most erosive effect (Wei et al., 2007); and the most severe annual erosion rates appeared in drought years, rather than in wet years (Wei et al., 2010); therefore more attention should be paid to rainfall variability and distribution to understand soil erosion processes in the LP. For vegetation patterns, shrubs are a superior choice to control soil erosion compared to other vegetation types (Wei et al., 2007; Zhou et al., 2016), because of their differences in canopy structure and surface litter layer (Zhou et al., 2016). Comparative experiments confirmed that soil compaction processes occurred during revegetation, which resulted in the increase of runoff and non-significant decrease of soil erosion in runoff plots, even though vegetation increased (Liu et al., 2012). Furthermore, the combination of plant traits, such as a small mean root diameter and great root tensile strength, can exert the greatest soil erosion control effect (Zhu et al., 2015). Overall, the main drivers of soil erosion in the LP include the characteristics of precipitation (e.g., intensity and duration), vegetation patterns, soil properties, and vegetation traits, which could inspire effective practices to prevent soil erosion (Zhou et al., 2016).

Ecological restoration in the LP altered the carbon and water cycles, even creating a severe carbon-water conflict (Figure 3). After the implementation of the GTGP, the NPP (Net Primary Productivity) and NEP (Net Ecosystem Productivity) have steadily increased, and a total of 96.1 Tg of additional carbon was sequestered from 2000 to 2008, transforming the LP from a net carbon source to a net carbon sink (Feng et al., 2013). However, the introduced vegetation will inevitably consume a large amount of water resources, causing a conflict between plant growth and water consumption. Numerous field monitoring and experiments have reported that soil moisture decline and soil desiccation occurred in the LP due to vegetation restoration (Chen et al., 2010; Liang et al., 2018; Wang et al., 2013; Yao et al., 2012). In addition, the remote sensing and model-based approach reached similar conclusions, namely that overplanting is responsible for soil drying in the LP (Zhang et al., 2018). Based on currently revegetated areas and water demand for human consumption, Feng et al. estimated a NPP threshold of 400 $\pm$ 5 g C m$^{-2}$ yr$^{-1}$, above which the population will suffer water shortages, and the results obtained indicated that the NPP in these revegetated areas was close to this limit (Feng et al., 2016). Furthermore, Zhang et al. estimated the equilibrium vegetation cover over the LP based on an ecohydrological model and ecosystem optimality theory (i.e., minimum water stress assumption), and compared it with the actual vegetation cover obtained from remote sensing data (Zhang et al., 2018). The results showed that the current vegetation cover (0.48 on average) has already exceeded the climate-defined equilibrium vegetation cover (0.43 on average) in many parts of the LP, especially in the middle-to-east regions (Zhang et al., 2018). In addition to excessive vegetation density, the types of introduced vegetation can also have a great impact on soil drying. Field experiments showed that soil moisture conditions were better where native species, rather than exotic species, grew. For example, pine and *Robinia pseudoacacia* plantations may not be appropriate for ecological restoration in a semi-arid loess hilly area such as the LP, due to the high-water consumption and poor water retention of these plants (Chen et al., 2010; Liang et al., 2018). Even land abandonment is a more effective method to maintain soil water than re-vegetation with introduced plants (Yao et al., 2012; An et al., 2017).





Runoff and sediment transport have a great impact on biogeochemical processes that determine ecosystem health in river basins (Gao et al., 2016; Gao et al., 2017). Understanding the dominant mechanism behind the variation of flow-sediment relationships is crucial to ensure a sustainable water management (Gao et al., 2017). The LP used to contribute nearly 90% of the Yellow River sediment (Wang et al., 2017). However, a sharp reduction (58%) in the river sediment was detected after 1979 (Wang et al., 2017) (Figure 3). Based on the relationship between cumulative sediment load and cumulative precipitation

described in (Wang et al., 2016), this sediment reduction trend can be classified into three periods: 1951–1979 (1.35 ± 0.65 Gt/y), 1980–1999 (0.73 ± 0.28 Gt/y), and 2000–2010 (0.32 ± 0.24 Gt/y). To elucidate the complex driving mechanism of sediment load change, an attribution method was developed to analyze 60 years of runoff and sediment load observations from the traverse of the Yellow River over the LP (Wang et al., 2016). The results showed that human activities are the main cause of sediment reduction in the Yellow River. Specifically, before 2000, landscape engineering, terracing, and the construction

of check dams and reservoirs were the primary factors driving the sediment load reduction, accounting for 54%; while after 2000, the main contributor was vegetation restoration, accounting for 57% (Wang et al., 2016). Reduced runoff and sediment in the Yellow River can produce a series of ecological and socio-economic impacts. For instance, sediment reduction can improve the water quality, but it can also lead the shrinking of the Yellow River delta, and the inadequate environmental flows may affect aquatic habitats (Wu et al., 2020).

In summary, the studies conducted on soil erosion and water-carbon processes, and on flow-sediment changes caused by vegetation restoration in the LP (Feng et al., 2016; Gao et al., 2016; Wang et al., 2016; Zhou et al., 2016) are of great significance for the establishment of future vegetation restoration measures. However, there are still some limitations that should be considered in future investigations. First, the coupling level of multiple ecological processes considered in the current studies is still insufficient. For example, soil erosion is simultaneously affected by multiple natural and anthropogenic factors,

including precipitation, terrain, soil properties, land use types, and even vegetation root traits (Zhou et al., 2016; Zhu et al., 2015). The current studies mainly coupled only two or three processes, such as precipitation and land use types, to study soil erosion in the LP (Zhou et al., 2016), which may increase the uncertainty of the results. Second, a basin-wide perspective to study the eco-hydrological effects caused by ecological restoration in the LP is still lacking (Wang et al., 2017). The upper, middle, and lower reaches of the Yellow River are interrelated; and therefore, ecological restoration carried out in the LP area located in its middle reaches will inevitably influence also the areas in the upper and lower reaches. A typical problem that

attracted considerable attention is the shrinking of the Yellow River delta caused by the reduction of sediment derived from the LP (Wu et al., 2020).

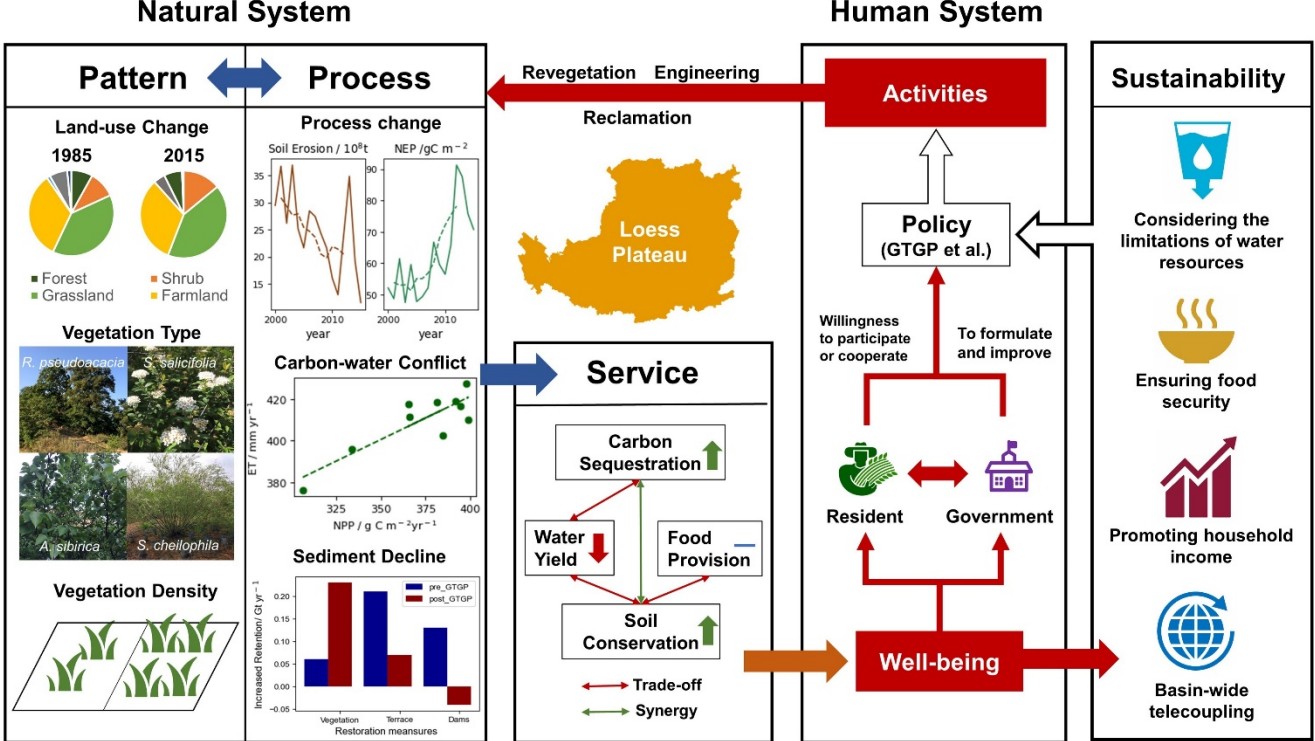

**Figure 3: Illustration of the coupled human and natural system studies of China's Loess Plateau using the "Pattern-Process-Service-Sustainability" framework.**

## 4.2 Linking ecosystem processes to ESs

Ecosystem composition, structure, and processes support the delivery of services from ecosystems to human society (Fu et al., 2013). By altering ecosystem patterns and processes, humanity can obtain desired services such as provision of food, freshwater, and timber, but this often leads to declines in some other services (Bennett et al., 2009). This trade-off effect is due to the complex relationship between ecosystem processes and ESs. Therefore, understanding the complex relationships between ESs requires linking ecosystem processes and ESs, which also could support optimized ecosystem management measures (e.g., minimize trade-offs and maximize synergies) and the sustainability of CHANSs (Fu et al., 2013).

Ecological restoration has changed the LP's land use, which will inevitably affect ESs (Figure 3). From 2000 to 2015, cultivated land in the LP decreased by 12.06%, while shrub land, grassland, artificial surfaces, and water bodies increased by 19.64%, 3.76%, 96.02%, and 20.21%, respectively (Liu et al., 2019). The long-term AVHRR NDVI average changed after 1999, the slope increased from 0.0018 y$^{-1}$ in the 1982–1998 period to 0.0053 y$^{-1}$ in 1999–2015 (Wu et al., 2019). The piecewise linear regression demonstrated that most breakpoints in the NDVI time series were detected between 1999 and 2002 (Wu et al., 2019). This means that, since the GTGP implementation in 1999, the LP has been greening.

By combining multi-source data, such as land use, remote sensing, and statistics, and by utilizing site-based monitoring data to set localized parameters, many ES assessments have been conducted in the LP at multiple scales (Lu et al., 2014; Zheng et



al., 2016; Su et al., 2020). Using the universal soil loss equation, the soil conservation service was quantified, and the results showed that total soil erosion in the LP significantly decreased by $0.96 \times 10^8$ t y$^{-1}$ between 2000 and 2015 (Wu et al., 2019). The sediment load between the Toudaotai and Tongguan stations decreased significantly by $0.25 \times 10^8$ t y$^{-1}$ during the same period (Wu et al., 2019). Accordingly, the nutrient (N and P) retention service increased in the areas subjected to revegetation

(Li et al., 2019). The carbon sequestration service, which was quantified by the terrestrial Carnegie-Ames-Stanford Approach (CASA) ecosystem model, showed a significant increasing trend. Estimates showed that the total regional NPP increased by $9.3 \pm 1.3$ g C m$^{-2}$ y$^{-1}$ between 2000 and 2010 (Feng et al., 2016). The spatial extent of the increased NPP and ET observed from MODIS (Moderate Resolution Imaging Spectroradiometer) coincides with that of GTGP implementation (Feng et al., 2016), indicating that vegetation restoration, rather than climate change, is the main cause of the estimated increase of annual

NPP or ET in the LP. However, water-related ESs, such as water supply and baseflow regulation, decreased dramatically at watershed (Luo et al., 2019) and regional scales (Lü et al., 2012; Wang et al., 2017). In contrast, the provision of services did not seem to be negatively affected by the reduction in cultivated land (Lü et al., 2012; Li et al., 2019), and the total grain output in the LP increased by 56.7% from 2000 to 2015, due to improved agricultural production techniques (Wu et al., 2019).

The assessment of ESs further provide the basis for the analysis of their trade-off and synergy relationships (Figure 3). Overall,

the same ES types (e.g., provisioning and regulating services) tended to show synergistic relationships, possibly due to the fact that they shared similar ecosystem processes and drivers (Bennett et al., 2009). For example, four provisioning services, namely grain production, oil crop production, livestock supply, and water supply, in the LP showed synergistic relationships, the exception being that between water supply and oil crop production (Li et al., 2019); and three regulating services, namely, carbon sequestration, soil conservation, and N and P retention, were positively correlated with each other (Li et al., 2019).

Although the negative impact of vegetation restoration on agricultural production is small, the trade-off between food supply and regulating services still exists in some local areas. The trade-off between water yield and soil conservation and the trade-off between water yield and carbon sequestration services have received more attention, because of the importance of water resources for the sustainability of the LP ecosystem (Jia et al., 2014; Su et al., 2020). In addition, there are scale effects on the trade-offs and synergies among ESs. For example, studies have reported that NPP and water yield, and sediment retention and

water yield showed trade-offs at the watershed scale, while they were not correlated at the LP scale (Su et al., 2020). Therefore, it is necessary to elucidate the relationships among ESs at multiple scales to promote their sustainable use and facilitate policy-making. Hu et al. developed a spatial assessment and optimization tool for regional ESs (SAORES) to evaluate and optimize regional water yield, soil erosion control, carbon sequestration, and food supply services in the LP (Hu et al., 2015). Based on multi-objective trade-offs among ESs, SAORES was able to provide suggestions for maximizing key ESs and optimizing land-

use patterns in the context of the GFGP (Hu et al., 2015).

In summary, the assessment of critical ESs and their trade-off, and synergy relationships in the LP provided theoretical support for the spatial optimization of ESs (Figure 3). However, there are still research gaps that should be considered in future investigations. First, current studies mainly focused on the ES supply or capacity, i.e., the capacity of an ecosystem to provide services irrespective of its use by humans (Vallecillo et al., 2019), but the ES demands, i.e., the amount and type of services



required or desired by society (Zwierzchowska et al., 2018) or local residents, have been largely neglected (Dong et al., 2021). The residents of LP are stakeholders with an interest in ESs and ecosystem management, and neglecting their demand and well-being may threaten the sustainability of ecological restoration outcomes. For example, studies have reported that a growing number of local farmers have left their homeland to become migrant workers, which may threaten the sustainable social and economic development of local villages (Dong et al., 2021). Second, current studies mainly consist of place-based

assessments of ESs, largely neglecting the flow of ESs between regions. Distant regions are connected by a process called telecoupling, which means that the use of ESs in one region may be affected by the management of ESs in other locations (Koellner et al., 2019). Therefore, scientists and decision makers need to focus more on the changes of demand and wellbeing of in the LP following ecological restoration, and on the flow and transfer of ESs between regions.

**4.3 Social-ecological sustainability**

Based on a deep understanding of the ecohydrological effects caused by vegetation restoration, changes in critical ESs and their trade-off, and synergy relationships (Figure 3), the social-ecological sustainability in the LP can be enhanced through targeted policies and management.

Enhancing water security. The LP is mainly located in arid and semi-arid regions, and water security is the priority issue that should be considered by scientists and policy-makers (Lu et al., 2014). Revegetation in the LP has well controlled soil erosion

and enhanced carbon sequestration, but it has simultaneously created a carbon-water conflict (Feng et al., 2016). Due to high density planting, introduced exotic plant species, and mismanagement of planted vegetation, large areas subjected to revegetation suffered soil drying (Chen et al., 2015; Wang et al., 2011a). Studies have shown that the current vegetation cover has already exceeded the climate-defined equilibrium vegetation cover in many parts of the LP (Zhang et al., 2018). Considering the minimum water needed for socioeconomic activities, local revegetation in the plateau is approaching

sustainable water resource limits (Feng et al., 2016), threatening the sustainability of the CHANS. In response to this issue, some overplanted areas may need to be reasonably thinned to reduce vegetation density. In addition, a better fit of plant species and planting density to the LP's natural environment, as well as an adaptive management of revegetation areas, are needed in the ensuing ecological restoration process (Lü et al., 2012; Feng et al., 2016).

Enhancing food security. Although the total grain production in the LP increased because of the development of agricultural

facilities and application of modern technologies (Wu et al., 2019), some counties still faced a decrease in grain production, and further revegetation may threaten the local food supply (Chen et al., 2015). To obtain a higher yield from the declining cropland areas, local people tended to increase the use of agrochemicals, in order to maintain or enhance land productivity, and ensure food security (Wang et al., 2014a). As a result, diffused pollution from agriculture has increased, and the quality of local land, groundwater, and surface water have been affected (Wang et al., 2014a). In the LP, check dams can curb

sediments flowing into the Yellow River, serve as carbon storage and sequestration structures, and they can also create a large number of high quality croplands when they are filled with sediments (Wang et al., 2014b; Wang et al., 2011b). Monitoring data have shown that dam croplands have a higher agricultural productivity than terrace and slope croplands have (Wang et





al., 2011b). However, check dams are at risk of collapse during rainstorms, due to the lack of management. Therefore, to achieve food security, relevant stakeholders should better manage the existing croplands, adopt advanced agricultural
production techniques to improve food yield per unit area, and simultaneously avoid agricultural pollution. In addition, more new croplands can be created by filling gullies, where conditions allow it (Liu et al., 2013b; Wang et al., 2018a).

Implementing basin-wide management. Scientists and policy makers need to consider not only the local ecohydrological effects caused by vegetation restoration, but also its effects on distant regions through telecoupling processes (Wu et al., 2020). In the past, the expansion of cultivation and deforestation to meet local food demand caused environmental degradation and
severe soil erosion in the LP (Fu et al., 2017). The sediments produced flowed into the Yellow River, leading to the rising of riverbed levels (Chen et al., 2015) and extension of the Yellow River delta (Kong et al., 2015). After ecological restoration, the severe soil erosion decreased and the environment improved in the LP, but the subsequent reduction of runoff and sediment load in the lower Yellow River, together with the change of water-sediment dynamics, generated cross-scale effects (Wu et al., 2020). The Yellow River delta has shifted to an erosional phase (Bi et al., 2014), which might affect more than two million
people and the biodiversity in distant, but coupled, ecosystems (Zhou et al., 2015). Therefore, a basin-wide perspective is needed to revisit the ecohydrological effects of ecological restoration. In addition, the establishment of a basin-wide ecosystem and land use management regime is needed to support sustainable water use and sediment regulation (Wang et al., 2017; Wang et al., 2016; Zhou et al., 2015).

Toward a socioeconomic development with low ecological impact. In addition to the environmental improvement, the GTGP
in the LP had also positive socioeconomic effects (Wu et al., 2019; Wu et al., 2021b). The program converted numerous farmlands to forest or grassland, which released local rural labor from crop production and promoted the transformation to non-farm activities (Liu et al., 2008; Uchida et al., 2009). A comparison between participants and nonparticipants in the GTGP using survey data on rural household livelihoods showed that the program improved rural household income, while simultaneously decreasing income inequality (Li et al., 2011). In parallel to the GTGP, multiple socioeconomic and political
factors, such as economic development, industrialization, and urbanization, also influence the livelihoods and income of rural households (Bryan et al., 2018; Ouyang et al., 2016). A comprehensive analysis that integrated metacoupled factors, i.e., human-nature interactions at different distances (Liu, 2017), proved that local economy and investment played dominant roles in the improvement of rural household income in the LP, while tourism, investment, and rural migration could boost non-farm work participation and effectively use the surplus laborers released from the GTGP, indirectly increasing income (Wu et al.,
2021b). To promote socioeconomic development in the LP and enhance the benefits that participants gain from non-farm work, it is necessary to promote initiatives that can create more local non-farm employment and help non-farm participation, such as promoting tourism industry, attracting more visitors and investment from outside, and providing training for new earning skills (Wu et al., 2021b; Cao, 2011; Yin et al., 2014). Meanwhile, barriers for rural labor migration should be overcome by offering equal opportunities and information services to migrant workers (Yang et al., 2018), especially to rural laborers that have less
local employment opportunities and wages.



Maintaining the GTGP achievements. The sustainability of the conservation achievements obtained through the GTGP is affected by the post-program land use plans of participants (Page and Bellotti, 2015; Deng et al., 2016). A study of the effects of regional ES changes and individual characteristics on participants' post-program land reconversion willingness in a watershed of the LP has shown that participants with a higher household income, more household employment change, higher

household involvement in crop production, and higher ecological awareness tended to not reconvert their GTGP land to agriculture use (Wu et al., 2021a). At the regional level, the changes of water yield and grain production services would affect the reconversion willingness of participants with different individual characteristics (Wu et al., 2021a). With a more comprehensive understanding of the effects of multiple-level factors on the reconversion willingness, the sustainability of the GTGP achievements can be enhanced by diversifying and improving the household income of participants (Bryan et al., 2018),

enhancing the participants' ecological awareness and recognition of ecological benefits obtained from the program (Song et al., 2014), and providing place-based solutions and protection priorities of ESs (Wu et al., 2021a).

## 5 Future directions of the Pattern-Process-Service-Sustainability paradigm

In this review, the LP was used as an example to illustrate the adoption of the "Pattern-Process-Service-Sustainability" paradigm to analyze the dynamics of CHANS and identify management priorities to enhance sustainability. After the

implementation of the GTGP in 1999, the LP has achieved the general "win-win" gains of restoring the environment and promoting socioeconomic development (Wu et al., 2019). However, water limits and land shortages are becoming the boundary of regional development (Wang et al., 2018a). It is necessary to readjust the revegetation strategy and promote socioeconomic development and livelihood transition, and consequently, the social-ecological sustainability of the LP (Wang et al., 2018a; Li et al., 2017). By revealing of the limitations of current studies, future research using the "Pattern-Process-Service-

Sustainability" paradigm to observe, analyze, and predict the dynamics and changes of CHANSs should especially focus on the following aspects:

(1) Integrated research on multiple processes. CHANSs are complex systems in which multiple natural and social processes interact and co-evolve (Quintas-Soriano et al., 2021), and therefore only focusing on one, or a few, of them may not reveal the essential laws behind complex phenomena (Fu et al., 2019). Multiscale systematic research quantifying the interactions among

the hydrological, pedologic, atmospheric, biological, and socioeconomic processes is necessary to provide integrative and convincing guidance to achieve sustainability (Fu, 2020). The driving mechanisms behind these interactions should also be further studied. This proposed research can specifically investigate the interactions among water, soil, atmosphere, and ecosystem, and their ecological effects; the biogeochemical processes and mechanisms on the earth's surface; and the interactions of different earth spheres and their response mechanisms to global change (Fu, 2020).

(2) Cascades of ecosystem structure, function, services, and human-wellbeing. As a link between nature and human society, the ES concept has become an important component of the scientific discourse, and a research object in science and policy (Carpenter et al., 2009). Understanding the ES delivery process from ecosystems to human society is essential to effectively



manage ecosystems (Mandle et al., 2021). The research frontiers of ES cascades involve the identification of the spatial mismatch between ES supply and demand, the differentiation of ES supply and human use, and the analysis of interregional
ES flows to support cross-regional management cooperation (Koellner et al., 2019; Vallecillo et al., 2019; Zwierzchowska et al., 2018). In addition, the relationship between ESs and human-well-being has also become an important research topic (Mandle et al., 2021). Further research needs to explore the coupling mechanisms of ES cascade components, clarify the dominant mechanisms of ES trade-offs to enhance and optimize ESs for the achievement of regional ecological security.

(3) Feedback mechanisms of human and natural systems. Human and natural systems interact and co-evolve over time and
have substantial impacts upon each other, with causality operating in both directions (Fu and Li, 2016; Quintas-Soriano et al., 2021). Understanding the complex interactions and feedbacks of CHANSs is one of the core research objectives in the field of sustainability. Future research needs to examine the bidirectional feedbacks between human and natural systems, the dynamics and resilience of CHANSs, the interactions between humans and nature (i.e., telecoupling) at multiple scales, the structural and dynamic fit of human and natural systems, and the capacity boundaries of CHANSs (Wang et al., 2018b).

(4) Data and models for sustainability. Field monitoring, control experiments, and remote sensing of natural systems at multiple scales provide abundant datasets to understand ecological processes, while the development of information technology and big data can help detect human activities and strengthen the spatiotemporal links between socioeconomic and natural processes. However, due to the scale mismatch between natural and social-economic processes, data assimilation is needed to integrate data from different scales, disciplines, and sources, and form a CHANS dataset for further sustainability analyses (Fu, 2020).
In addition to the system dataset, the development of a model, or model systems, is crucial for an integrated CHANS research. This kind of model can be realized through the integration of multiple ecological, human, and socioeconomic models (Fu and Li, 2016).

## 6 Conclusion

We proposed a conceptual cascade framework of "Pattern-Process-Service-Sustainability" to observe, analyze, and predict the
dynamics of CHANSs and support the design of policies and measures that promote sustainability. To illustrate the use of this framework, this study systematically examined the CHANS research experiences in China's LP, in terms of coupling landscape patterns and ecological processes, linking ecological processes to services, and promoting social-ecological sustainability. Since 2000, the vegetation coverage in the LP has increased due to ecological restoration. Soil erosion has been well controlled, and the sediment derived from the LP, and flowing into the Yellow River, has also greatly decreased; however, overplanting,
the introduction of exotic plant species, and the mismanagement of planted vegetation have also led to soil drying in regions subjected to revegetation. Ecosystem services, especially for soil conservation and carbon sequestration, have significantly improved, but a trade-off between carbon sequestration and water supply has been identified at multiple scales. Some social-ecological issues, such as water security, food security, and the shrinking of the Yellow River delta, have emerged in the LP, posing a threat to its sustainable development in the future. To promote sustainability, scientists and policy-makers should pay



more attention to the socioeconomic dimensions of the LP and apply a basin-wide and telecoupling perspective to formulate land use policies, and carry out ecological restoration more effectively in the future.

Considering the typicality of the social-ecological context in the LP, we believed that research experiences from this region are relevant for CHANS studies in the rest of the world. Future research using the "Pattern-Process-Service-Sustainability" paradigm should especially examine the integrated research on multiple processes, the cascades of ecosystem structure,

function, services, and human-wellbeing, the feedback mechanisms of human and natural systems, and the data and models for sustainability.

## Data availability

No data sets were used in this article.

## Author contributions

B.F. designed the research. Xutong Wu, Z.W., and Xilin Wu performed the literature review. B.F., Xutong Wu, Z.W., S.W., and Xilin Wu contributed to the interpretation and writing.

## Competing interests

The authors declare no competing interests.

## Acknowledgements

This research was financially supported by the National Natural Science Foundation of China (41930649), China National Postdoctoral Program for Innovative Talents (BX2021042), and China Postdoctoral Science Foundation (2021M700458).

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
