# Peer review of "Coupling human and natural systems for sustainability: experiences from China's Loess Plateau"

_Earth System Dynamics, 2022_

## Author Response (AR1)

**Response to Reviews (Ms. No. esd-2022-1)**

**To Reviewer #1:**

Thank you for your valuable suggestions and the time that you have spent reviewing our manuscript. We have followed all your suggestions in our revision. Below are the reviewer's previous comments, followed by our responses.

Sincerely,

Bojie Fu (on behalf of the author team)

\*\*\*\*\*\*\*\*\*\*

Reviewer #1 (General comments):

This paper provides a comprehensive overview of the conceptual cascade framework of "Pattern-Process-Service- Sustainability", and also provides a concrete example on the application of this framework at the Loess Plateau in China. This region has been subject to major vegetation restoration activities during several decades, and the impacts of these activities have been extensively studied by different methodologies. This accumulated research provides an excellent example of the value of systematic and integrated studies for solving socio-ecological problems on the very large scale.

I do not have any major comments on the methodologies or content, because the study mainly reviews results of already published material in well-known journals and puts them into the conceptual framework. The authors clearly have a very good knowledge about the recent literature in the field, and the context of the research.

**[Response]** We thank you for your positive comment.

Specific comments:

**[Reviewer #1 Comment 1]** lines 66-58 I would suggest to use present tense (i.e. "is introduced" and "is systematically illustrated", because this refers to aims of the present paper. I would suggest checking this also in other places (e.g. Conclusions).

**[Response]** Thank you for your suggestion. Following your suggestion, we revised the sentence as: "This paper starts with a detailed introduction of the proposed framework and its components, then systematically illustrates the use of this framework to study the dynamics of CHANSs and support decision making for the promotion of sustainability through a review of research experiences in China's Loess Plateau (LP)." (Line 66-68). In addition, we carefully checked the tense of the whole manuscript.

**[Reviewer #1 Comment 2]** Section 3 China's Loess Plateau: I would propose to include a table giving some more details about the area, such as T, soil and vegetation types, population, etc.

**[Response]** According to your suggestion, we added a table to give more details about the Loess Plateau, including annual mean temperature, precipitation, and potential evapotranspiration, soil and vegetation types, geomorphology, population, and some other socioeconomic characteristics.

Added table:

**Table 1 Regional characteristics of the LP** (Fu et al., 2017; He et al., 2021)

| Natural characteristics | | Socioeconomic characteristics | |
|---|---|---|---|
| Annual mean temperature | 8-14 ℃ | Population | Increased from 42 million in 1950 to 109 million in 2019 |
| Annual mean precipitation | ~ 400 mm | Per capita GDP | 7380 USD (2019) |
| Annual mean potential evapotranspiration | > 1000 mm | Urbanization rate | ~ 56% (2019) |
| Soil type | Los orthic entisols, dark and soild-cumulic ustic isohumisols, ustic lubisols, etc. | Main crop | Wheat and maize |
| Vegetation type | From nowthwest to southeast: desert, desert-steppe, steppe, forest-steppe, and forest | Grain self-sufficiency rate | Incrased from 62% in 1950 to 108% in 2019 |
| Geomorphology | Loess Yuan, Liang, Mao, and various valleys of different erosion magnitude | Ecological restoration project | Three-North Shelterbelt Development program (1978), Soil and Water Conservation Program (1983), and Grain for Green program (1999) |

**[Reviewer #1 Comment 3]** lines 175-180 Is there a gradient in precipitation that would affect this NPP limit given and has there been any changes in precipitation during this period? Is climate change likely to cause any further water stress in this region that could affect these conclusions? These topics would require a little more discussion here.

**[Response]** Thank you for your comment. There is a precipitation gradient from the south-east to the north-west in the LP, and the precipitation of the LP showed no significant change from 2000 to 2015 (Wu et al., 2019). In Feng et al.'s study, the estimated NPP threshold is the annual NPP at the LP scale, and therefore the effect of different precipitation gradients on the permissible NPP threshold is not considered (Feng et.al, 2016). We added more discussions on future climate change and its effects on this permissible NPP limit: "Future climate change will affect this permissible NPP threshold. Considering the maximum, median, and minimum scenarios of precipitation change and future changes in plant water-use efficiency, permissible NPP by 2050 is $578 \pm 48$ g C m$^{-2}$ yr$^{-1}$, $473 \pm 41$ g C m$^{-2}$ yr$^{-1}$, and $309 \pm 29$ g C m$^{-2}$ yr$^{-1}$, respectively (Feng et al., 2016)." (Line 184-187)

**References:**

Feng, X., Fu, B., Piao, S., Wang, S., Ciais, P., Zeng, Z., Lü, Y., Zeng, Y., Li, Y., Jiang, X., and Wu, B.: Revegetation in China's Loess Plateau is approaching sustainable water resource limits, Nature Climate Change, 6, 1019-1022, 10.1038/nclimate3092, 2016.

Wu, X., Wang, S., Fu, B., Feng, X., and Chen, Y.: Socio-ecological changes on the Loess Plateau of China after Grain to Green Program, Science of The Total Environment, 678, 565-573, 10.1016/j.scitotenv.2019.05.022, 2019.

**[Reviewer #1 Comment 4]** lines 220-228 There is some repetition of text here vs. the Introduction, please check.

**[Response]** Thank you for your reminding. We revised the beginning of Section 4.2 as: "By

70 coupling patterns and processes, researchers found that the ecological restoration program has altered the soil erosion and water-carbon processes and flow-sediment relationships in the LP (Feng et al., 2016; Wang et al., 2016). As ecological processes underpin the delivery of ESs, changes in these natural processes in the LP will affect ESs that local residents depend on (Fu et al., 2013). Linking ecological processes to ESs can help understand the complex relationships among ESs and

75 support optimized ecosystem management measures (e.g., minimize trade-offs and maximize synergies) and sustainability of CHANSs (Fu et al., 2013). Regarding this aspect, a large number of studies about land use and land cover change, ES assessments, and their trade-off and synergy analysis have been conducted in the LP over the past two decades." (Line 227-233)

80 **[Reviewer #1 Comment 5]** line 283 Use colon (:) after "Enhancing water security" or subheading here. The same goes for the other subsections ("Enhancing food security" and so on).
**[Response]** Following your suggestion, we used colon (:) after the four subtitles.

**[Reviewer #1 Comment 6]** line 403-406 Future directions: I would assume that climate change

85 impacts on NPP and water yield would be a key topic for future research?
**[Response]** Thank you for your suggestion. The future directions we proposed in Section 5 "Future directions of the Pattern-Process-Service-Sustainability paradigm" are oriented to the framework and CHANS studies, and therefore they are comparatively macroscopic. The future research direction "Climate change impacts on NPP and water yield" is a very valuable and specific research

90 topic in the LP. Therefore, we added this in Section 4.3 "Social-ecological sustainability": "Besides, it is necessary to consider the impact of climate change on NPP and water yield to meet future challenges of water shortage." (Line 314-315)

**[Reviewer #1 Comment 7]** line 550 This reference seems to be incomplete: Ouyang, Z., H, Z., Y,

95 X., S, P., J, L., W, X., Q, W., L, Z., Y, X., and E, R.: Improvements………
**[Response]** We revised this reference as "Ouyang, Z., Zheng, H., Xiao, Y., Polasky, S., Liu, J., Xu, W., Wang, Q., Zhang, L., Xiao, Y., Rao, E., Jiang, L., Lu, F., Wang, X., Yang, G., Gong, S., Wu, B., Zeng, Y., Yang, W., and Daily, G. C.: Improvements in ecosystem services from investments in natural capital, Science, 352, 1455-1459, doi:10.1126/science.aaf2295, 2016." (Line 571-573)

100

In conclusion, I think this is a valuable paper that deserves to be published after making these rather minor additions/changes suggested.
**[Response]** We thank you for your positive comments and the time that you have spent reviewing our manuscript.

105

**To Reviewer #2:**

Thank you for your valuable suggestions and comments. We have revised our manuscript following all your suggestions. The detailed responses are as follows.

110

Sincerely,

Bojie Fu (on behalf of the author team)

\*\*\*\*\*\*\*\*\*\*\*

115 Reviewer #2 (General comments):

Through a thorough review of current research on coupled human and natural system (CHANS), the manuscript proposes a conceptual cascade framework of "Pattern-Process-Service Sustainability" and systematically demonstrates its applications in China's Loess Plateau (LP) in terms of coupling

120 landscape patterns and ecological processes, linking ecological processes to services, and promoting social-ecological sustainability. By identifying the current research limitations, it points out the need for " the integrated research on multiple processes, the cascades of ecosystem structure, function, services, and human-wellbeing, the feedback mechanisms of human and natural systems, and the data and models for sustainability", especially implementing basin-wide management to consider

125 not only the local ecohydrological effects caused by vegetation restoration, but also its effects on distant regions through telecoupling processes. The manuscript is well-written and innovative. It provides a much needed CHANS framework to address the bottleneck challenges facing both research and policy communities. The manuscript is of interest not only to the readers of the journal but also the wider ecological, socioeconomic, humanity, and management communities. I only have

130 the following minor suggestions for the authors to consider:
**[Response]** We thank you for your positive comment.

**[Reviewer #2 Comment 1]** 1a shows the human system and natural system affects sustainability in one singular direction, while Fig.3 depicts sustainability impacts the interactions of both human and

135 natural systems. Thus, Fig.1a should be modified to be consistent with Fig.3.
**[Response]** Thank you for your suggestion. We added an arrow from "Sustainability" to "Human and natural systems" in Fig. 1 and ensure that Fig.1 and Fig. 3 are consistent.

Revised figure:

[Figure]

140

**Figure 1: Illustration of the "Pattern-Process-Service-Sustainability" framework.** Pattern, process, service, and sustainability in the coupled human and natural system (left) and diagram of typical patterns, processes (e.g., hydrological processes) and ESs (e.g., provision services) (right).

145 **[Reviewer #2 Comment 2]** 3 depicts both trade-offs and synergy but appears emphasizes trade-offs and synergy only occurs between ecosystem services. Would defining, implementing/achieving sustainability by human activity also lead to synergy between enhancing the capacity of natural system and the well-being of human being across multiple scales as well?

**[Response]** Thank you for your comment. Indeed, this study emphasizes the trade-offs and

150 synergies between ecosystem services. We argue that implementing sustainability activities such as land use optimization is an effective way to mitigate trade-offs and enhance synergies, which finally can achieve win-win gains between people and nature (i.e., the synergy between natural systems and human well-being). This is reflected by the two inverted arrows from "Sustainability" to "Human systems (Activities)" then to "Natural system" in Figure 3. In addition, we added a sentence

155 in Section 2 "The Pattern-Process-Service-Sustainability framework": "Targeted sustainability activities such as land use optimization provide feasible and effective ways to manage landscape patterns and ecosystem processes and to mitigate trade-offs and enhance synergies among ESs, which finally bring win-win gains between human and nature and improve the sustainability of CHANSs." (Line 103-106)

160

**[Reviewer #2 Comment 3]** Based on the research experiences in the LP, the manuscript suggests future scientists and policy-makers "apply a basin-wide and telecoupling perspective to formulate land use policies, and carry out ecological restoration more effectively in the future." This key and also challenging recommendation applies not only to China's LP but also to the rest of the world. I

165 wonder if the authors can elaborate a little further on how to implement this suggestion from an institutional perspective.

**[Response]** Following your suggestion, we added more elaboration in Section 4.3: "The

establishment of a basin-wide ecosystem and land use management regime is needed to support sustainable water use and sediment regulation (Zhou et al., 2015; Wang et al., 2016; Wang et al., 2017). The interconnected sub-hydrological units of the Yellow River Basin span various human-defined boundaries and are managed by different agents, resulting in institutional fragmentation. Considering the holistic nature of the basin, policy-makers and managers of the LP should cooperate and coordinate with middle-stream and downstream stakeholders to integrate management of river water and sediment (Wang et al., 2019). Specifically, cross-border and cross-scale coordination exerted by a higher-level authority is an effective means to overcome institutional fragmentation. This is because a third party or a higher administrative agency with whole-basin responsibility can promote effective coordination on a basin level by establishing social ties indirectly linking actors across administrative levels (Wang et al., 2019)." (Line 321-329)

After these minor revisions, I recommend the manuscript be accepted for publication.

**[Response]** We thank you for your positive comments and the time that you have spent reviewing our manuscript.

185

Thank you for your valuable suggestions and the time that you have spent reviewing our manuscript. We have revised our manuscript according to your suggestions. Below are the reviewer's previous comments, followed by our responses.

190  Sincerely,

Bojie Fu (on behalf of the author team)

\*\*\*\*\*\*\*\*\*\*

Reviewer #3 (General comments):

195

The manuscript entitled "Coupling Human and Natural Systems for Sustainability: Experiences from China's Loess Plateau" proposed a conceptual cascade framework of "Pattern-Process-Service Sustainability" to observe, analyze and predict the dynamics of coupling human and natural system (CHANSs). The manuscript introduced the components of the framework in length and illustrated

200  a review of CHANS research in terms of coupling landscape patterns and ecological processes, linking ecological processes to services, and promoting social-ecological sustainability.

It is well known that Loess Plateau (LP) in China suffered severe soil erosion historically and achieved successful vegetation restoration in recent decades, and extensive studies have been conducted on the CHANS of the LP, so it is an ideal region for summarizing research experiences

205  of CHANS.

Overall, this is a well written manuscript without any apparent flaws. The article tackles a relevant topic, certainly of interest to the readership of the journal. However, some details and aspects are not clearly expressed. A list of the main issues is provided below in a constructive spirit. I also include a (non-exaustive) list of style suggestions and grammar issues. I recommend minor revision.

210  **[Response]** We thank you for your positive comment.

Some revisions to manuscript are needed:

Abstract:

**[Reviewer #3 Comment 1]** This manuscript focused on the conceptual cascade framework of

215  "Pattern-Process-Service-Sustainability" to understand the dynamics of CHANS, but the meaning of this framework is missing.

**[Response]** Thank you for your comment. We added the meaning of this framework in the Abstract: "To help understand the dynamics of a coupled human and natural system (CHANS) and support the design of policies and measures that promote sustainability, we propose a conceptual cascade

220  framework of "Pattern-Process-Service-Sustainability", which is characterized by coupling landscape patterns and ecological processes, linking ecological processes to ecosystem services, and promoting social-ecological sustainability." (Line 10-13)

**[Reviewer #3 Comment 2]** I thought the novelty of this manuscript is the proposed framework, but

225  in LL.70-71, it described that "Over the last 20 years, researchers have conducted extensive studies on the CHANS of the LP using the 'Pattern-Process-Service-Sustainability' framework", then what

is the originality of this manuscript? It should be clearly stated.

**[Response]** Thank you for your comment. Indeed, the novelty of this study is the proposed framework, which can help researchers better observe, analyze, and understand the dynamics of CHANS and support the design of policies and measures that promote sustainability. Previous studies in the LP were not guided by this framework and most of them separately focused on the landscape patterns, ecological processes, or ecosystem services. Our paper proposes this framework by summarizing research experiences from these studies in the LP. We revised the sentence to emphasize the novelty of this study: "Over the last 20 years, researchers have conducted extensive studies on the landscape patterns, ecological processes, and ESs of the LP, making it an ideal region for summarizing research experiences and providing guidance for the CHANS studies." (Line 68-70)

Introduction:

**[Reviewer #3 Comment 3]** The argument of proposed framework is well established, but the knowledge gap of current understanding towards CHANS is missing.

**[Response]** Following your suggestion, we summarized the knowledge gap of current understanding as: "As CHANSs involve multiple natural and human processes interacting at different scales, this complexity poses a challenge for researchers to understand their dynamics (Liu et al., 2007; Nelson et al., 2007; Rocha et al., 2015; Gunderson et al., 2017). However, research paradigms that explicitly guide sustainability practices based on deepened understandings of the interactions between humans and nature are still limited." (Line 47-50)

**[Reviewer #3 Comment 4]** The explicit aim of this research is also missing.

**[Response]** According to your suggestion, we added the aim of this research as: "To fill this knowledge gap, we propose a conceptual cascade framework of "Pattern-Process-Service-Sustainability" (Fu and Wei, 2018) to better observe, analyse, and understand the dynamics of CHANSs and promote their sustainability." (Line 50-52)

China's Loess Plateau:

**[Reviewer #3 Comment 5]** The success of the GTGP is shown in Figure 2 with NDVI index, but in the context, there is no description of NDVI, neither an explanation of the Figure 2 indicates.

**[Response]** Following your suggestion, we added several sentences in Section 3: "Since the beginning of the 21$^{st}$ century, the vegetation coverage in the LP has significantly increased and soil erosion has been effectively controlled (Fu et al., 2011b). According to the MODIS NDVI data, more than 70% of the LP showed a significant "greening" trend from 2000 to 2015, mainly distributed in the central and southern parts (Fig. 2)." (Line 138-140)

Application of the framework in the LP:

**[Reviewer #3 Comment 6]** The farmland in land-use changes is a type of also human activities, why it is divided into natural system?

**[Response]** Thank you for your comment. The phrase "land-use changes" more highlights the land types conversion caused by human activities. We changed it to "land use and land cover" (LULC), which is more related to the spatial pattern or configuration of natural systems. Besides, the phrase LULC is more neutral, and human land use such as cultivated land is a part of LULC.

**[Reviewer #3 Comment 7]** The human system and the linkage between natural system and ESs did not well depict in the context.

**[Response]** Following your suggestion, we revised the first paragraph in the Section 4.2 to illustrate the linkages between natural processes and ESs, and the impact of changes in ES on local residents: "By coupling patterns and processes, researchers found that the ecological restoration program has altered the soil erosion and water-carbon processes and flow-sediment relationships in the LP (Feng et al., 2016; Wang et al., 2016). As ecological processes underpin the delivery of ESs, changes in these natural processes in the LP will affect ESs that local residents depend on (Fu et al., 2013). Linking ecological processes to ESs can help understand the complex relationships among ESs and support optimized ecosystem management measures (e.g., minimize trade-offs and maximize synergies) and sustainability of CHANSs (Fu et al., 2013). Regarding this aspect, a large number of studies about land use and land cover change, ES assessments, and their trade-off and synergy analysis have been conducted in the LP over the past two decades." (Line 227-233)

**[Reviewer #3 Comment 8]** The introduction of social-ecological sustainability mainly focusses on the enhancement of targeted policies and management instead of the explanation and the coupling relationship of each component, it should be well-structured like 4.1 and 4.2.

**[Response]** Thank you for your comment. According to the framework, Section 4.3 is to diagnose the causes of the unsustainable status of CHANSs, and support the design of policies and measures that promote sustainability. We further elaborated the meanings and aims of our framework, and explained the coupled relationships between system components before giving targeted suggestions.

We added these sentences in Section 2: "The practical implications of CHANSs studies are to promote the "Sustainability" of the coupled systems (Ostrom, 2009; Liu et al., 2013a). A deep understanding of "Pattern-Process-Service" interactions can help us figure out the causes of the unsustainable status of a CHANS and provide the scientific basis for designing policies or measures that promote its sustainability. Both environmental changes and human activities can affect the interactions between landscape patterns and ecological processes, and directly or indirectly affect the ESs upon which humans depend (Fu et al., 2019). Targeted sustainability activities such as land use optimization provide feasible and effective ways to manage landscape patterns and ecosystem processes and to mitigate trade-offs and enhance synergies among ESs, which finally bring win-win gains between human and nature and improve the sustainability of CHANSs." (Line 99-106)

We revised the beginning of Section 4.3 as: "Studies that coupled patterns and processes, and linked processes to services can reveal the interactions among social-ecological components, and help us figure out the causes of the unsustainable status of a CHANS and the corresponding solutions. Although the vegetation coverage has improved greatly and soil erosion has been well controlled in the LP after two decades of ecological restoration, some accompanied issues have started to threaten the long-term sustainability of the LP. The carbon-water conflict accompanied with revegetation is a typical example (Feng et al., 2016). Due to high-density planting, introduction of exotic plant species, and mismanagement of planted vegetation, large areas subjected to revegetation suffered soil drying (Chen et al., 2015; Wang et al., 2011). Studies have shown that the current vegetation cover has already exceeded the climate-defined equilibrium vegetation cover in many parts of the LP (Zhang et al., 2018). Considering the minimum water needed for socioeconomic activities, local revegetation in the plateau is approaching sustainable water resource limits (Feng et al., 2016),

threatening the sustainability of the CHANS. In addition, although the total grain production in the LP increased because of the development of agricultural facilities and application of modern technologies (Wu et al., 2019), some counties still faced a decrease in grain production due to the conversion of cropland to forest or grassland, and further revegetation may threaten the local food supply (Chen et al., 2015). To obtain a higher yield from the declining cropland areas, local people tended to increase the use of agrochemicals to maintain or enhance land productivity (Wang et al., 2014). As a result, diffused pollution from agriculture has increased, and the quality of local land, groundwater, and surface water have been affected (Wang et al., 2014). Another important issue is the spillover effects of the LP on the telecoupled regions. In the past, the expansion of cultivation and deforestation to meet local food demand caused environmental degradation and severe soil erosion in the LP (Fu et al., 2017). The sediments produced flowed into the Yellow River, leading to the rising of riverbed levels (Chen et al., 2015) and extension of the Yellow River delta (Kong et al., 2015). After ecological restoration, the improved environment and decreased soil erosion in the LP led to the subsequent reduction of sediment load and runoff in the lower Yellow River (Wu et al., 2020). The Yellow River delta has shifted to an erosional phase (Bi et al., 2014), which might affect more than two million people and the biodiversity in distant, but coupled, ecosystems (Zhou et al., 2015). Based on a deep understanding of the ecohydrological effects caused by vegetation restoration, changes in critical ESs and their trade-off and synergy relationships, the social-ecological sustainability in the LP can be enhanced through the following targeted policies and management (Figure 3)." (Line 282-306)

Future directions of the Pattern-Process-Service-Sustainability paradigm:

**[Reviewer #3 Comment 9]** The first aspect that future research should focus on (integrated research on multiple processes) is repetitive with the "future investigation" in 4.1 (LL.208-212)

**[Response]** Thank you for your reminding. The future directions of the Pattern-Process-Service-Sustainability framework are proposed partly based on the limitation of current studies in the LP (Section 4.1). Therefore, there may be some repetitive content in these sections. We revised the limitations in Section 4.1 as: "First, the coupling level of multiple ecological processes is still insufficient. For example, soil erosion is simultaneously affected by multiple natural and anthropogenic factors, including precipitation, terrain, soil properties, land use and land cover types, and even vegetation root traits (Zhou et al., 2016; Zhu et al., 2015). The current studies in the LP mainly coupled only two or three processes, such as precipitation and land use and land cover types (Zhou et al., 2016), which may increase the uncertainty of the results." (Line 214-218)

Conclusions:

**[Reviewer #3 Comment 10]** 393-396 is repetitive with the abstract, it's better to rephrase.

**[Response]** According to your suggestion, we revised these sentences as: "Since 2000, the vegetation coverage in the LP has increased and soil erosion has been well controlled due to ecological restoration. However, overplanting, the introduction of exotic plant species, and the mismanagement of planted vegetation have also led to soil drying in some regions subjected to revegetation, and a trade-off between carbon sequestration and water supply has been identified at multiple scales. Some social-ecological issues, such as water resource limit, local food scarcity, and negative spillover effects, have emerged in the LP, posing a threat to its sustainable development in the future. To promote social-ecological sustainability, scientists and policy makers should pay more

attention to water and food security, basin-wide governance, maintenance of ecological restoration achievements, and rural livelihood transition." (Line 406-412)

**[Reviewer #3 Comment 11]** The implication of "Pattern-Process-Service-Sustainability" paradigm should be well introduced.

**[Response]** Thank you for your suggestion. We added the implication of our framework in the Conclusion: "A deep understanding of the reciprocal effect between landscape patterns and ecological processes, and complex linkages between ecological processes and ESs that support human well-being is crucial for promoting social-ecological sustainability. The conceptual cascade framework of "Pattern-Process-Service-Sustainability" proposed in this review can help researchers observe, analyse, and understand the dynamics of CHANSs, diagnose the causes of the unsustainable status, and support the design of policies and measures that promote sustainability." (Line 399-403)

Minor points:

1. 9: "Addressing the sustainability challenges facing humanity" is a bit strange, "Addressing the sustainability challenges that humanity is facing" sounds better?

**[Response]** We revised the sentence accordingly.

2. 15-20 are the evaluation result of the proposed framework, but the linkage of these results is missing.

**[Response]** We revised theses sentences as: "Ecological restoration in the LP has greatly increased its vegetation coverage and controlled its soil erosion. However, some accompanied issues like soil drying in some areas due to the introduction of exotic plant species and the mismanagement of planted vegetation, and water use conflicts between vegetation and human caused by the trade-off between carbon sequestration and water supply, have started to threaten the long-term sustainability of the LP. Based on the comprehensive understanding of CHANS dynamics, the social-ecological sustainability of the LP can be improved through enhancing water and food security, implementing basin-wide governance, maintaining ecological restoration achievements, and promoting rural livelihood transition." (Line 15-21)

3. 14-15 is repetitive with LL. 68-69, at least one sentence should be deleted.

**[Response]** Following your suggestion, we deleted the sentence in the Introduction.

4. 84 the "wellbeing" should be changed as "well-being"

**[Response]** We revised the word in the whole manuscript accordingly.

5. LL.86-87 "the same single ES" should delete "single"

**[Response]** We deleted the word accordingly.

6. 130 "soil-and water- conservation" should be "soil and water conservation"

**[Response]** We revised the phrase accordingly.

7. L.140 "the extensive research here conducted" should be "the extensive researches conducted

here"

**[Response]** We revised the sentence accordingly.

405

8. L.148 "As human disturbance to .." lack of the subjective in this sentence.

**[Response]** Thank you for your suggestion. We revised this sentence as "However, deforestation and afforestation can alter this equilibrium and cause a series of ecohydrological effects that influence water, carbon, soil processes, and the overall ecosystem sustainability (Feng et al., 2016; Zhang et al., 2018)." (Line 153-155)

410

9. L.195 "described in (Wang et al., 2016)" should delete "described in"

**[Response]** We revised the sentence accordingly.

415  10. 236 "Using the universal soil loss equation", I guess it's USLE (A = R x K x L x S x C x P), it's better to show

**[Response]** We added this equation according to your suggestion.

11. L.249 "The assessment of ESs further provide" should be "provides"

420  **[Response]** We revised the sentence as: "The assessments of ESs further provide the basis for the analysis of their trade-off and synergy relationships (Figure 3)." (Line 256)

12. 277-278 "…the changes of demand and wellbeing of in the LP following ecological restoration" is not clear

425  **[Response]** We revised this sentence as: "Second, as distant regions are connected by telecoupling processes, the use of ESs in one region may be affected by the management of ESs in other locations (Koellner et al., 2019), however, current studies mainly focus on place-based assessments of ESs, largely neglecting the flow of ESs between regions." (Line 278-280)

430  13. 286 "introduced exotic plant species" is better changed as "introduction of exotic plant species"

**[Response]** We revised the phrase accordingly.

14. LL.292-293 "are needed in the ensuing …" should be changed as "are needed for.."

**[Response]** We revised the sentence accordingly.

435

15. 302 "than terrace and slope croplands have" should be "than terrace and slope croplands do"

**[Response]** Thank you for your suggestion, this sentence was removed due to content adjustment.

16. 303 "check dams are at risk of collapse during rainstorms, due to the lack of management" please

440  add references.

**[Response]** Thank you for your suggestion, this sentence was removed due to content adjustment.

17. 320 "had also positive" should be "also had"

**[Response]** Thank you for your suggestion, this sentence was removed due to content adjustment.

445

18. L.365 "human-wellbeing" should be "human well-being"

**[Response]** We revised the word in the whole manuscript accordingly.

19. 380 "field monitoring, control experiments… to understand ecological processes" should add reference

**[Response]** We revised the sentence as: "Field monitoring, control experiments, and remote sensing of natural systems at multiple scales provide abundant datasets to understand ecological processes (Wei et al., 2010; Feng et al., 2010; Yao et al., 2012; Zhou et al., 2016; Liang et al., 2018), while the development of information technology and big data can help detect human activities and strengthen the spatiotemporal links between socioeconomic and natural processes." (Line 390-393)

20. Figure 1: The green color for describing "pattern" is improper in (b), or the transparency of background color is needed. The "sustainability" in the framework also can be shown in diagram (b).

**[Response]** Thank you for your suggestion. We changed the font color in 1b. In addition, we merged 1a and 1b into one figure, and add an arrow from "Sustainability" to "Human and natural systems" in Figure 1.

Revised figure:

[Figure]

**Figure 1: Illustration of the "Pattern-Process-Service-Sustainability" framework.** Pattern, process, service, and sustainability in the coupled human and natural system (left) and diagram of typical patterns, processes (e.g., hydrological processes) and ESs (e.g., provision services) (right).

21. Figure 2: The label of "Yellow River" is barely seen, the color of the label should be changed. What does the white color (the western LP) indicate regrading NDVI index?

**[Response]** Thank you for your suggestion. We changed the label of "Yellow River" to dark blue. The white color represents the non-significant trend of NDVI index. We added this in legend.

475 Revised figure:

[Figure]

**Figure 2: Location of the Loess Plateau and its normalized-difference vegetation index (NDVI) trend from 2000 to 2015.**

480 22. Figure 3: The descriptions of vegetation type is barely seen in the Figure, maybe change another color? The soil erosion, NEP change and the sediment decline are all process changes, so the subtitle of the "process change" is inappropriate

**[Response]** Thank you for your suggestion. We moved the vegetation name to the top of the image. In addition, we deleted "process change", and named the other two processes as "Vegetation
485 dynamics" and "Soil erosion".

Revised figure:

[Figure]

**Figure 3: Illustration of the coupled human and natural system studies of China's Loess Plateau using the "Pattern-Process-Service-Sustainability" framework.** Data used in this figure were obtained from (Wang et al., 2016; Feng et al., 2016; Liu et al., 2019; Wu et al., 2019).

490